# Evaluation of variation in special educational needs provision and its impact on health and education using administrative records for England: umbrella protocol for a mixed-methods research programme

Ania Zylbersztejn ,[1] Kate Lewis,[1] Vincent Nguyen,[1] Jacob Matthews,[2] Isaac Winterburn,[2] Lucy Karwatowska ,[1] Sarah Barnes,[2] Matthew Lilliman,[1] Jennifer Saxton,[2] Antony Stone,[1] Kate Boddy ,[3] Johnny Downs ,[4,5] Stuart Logan,[6] Jugnoo Rahi,[1,7] Kristine Black-Hawkins,[8] Lorraine Dearden,[9] Tamsin Ford ,[2] Katie Harron ,[1] Bianca De Stavola,[1] Ruth Gilbert[1]

**Correspondence to**
Professor Ruth Gilbert;
r.gilbert@ucl.ac.uk

## ABSTRACT

**Introduction** One-third of children in England have special educational needs (SEN) provision recorded during their school career. The proportion of children with SEN provision varies between schools and demographic groups, which may reflect variation in need, inequitable provision and/or systemic factors. There is scant evidence on whether SEN provision improves health and education outcomes.

**Methods** The Health Outcomes of young People in Education (HOPE) research programme uses administrative data from the Education and Child Health Insights from Linked Data—ECHILD—which contains data from all state schools, and contacts with National Health Service hospitals in England, to explore variation in SEN provision and its impact on health and education outcomes. This umbrella protocol sets out analyses across four work packages (WP). WP1 defined a range of 'health phenotypes', that is health conditions expected to need SEN provision in primary school. Next, we describe health and education outcomes (WP1) and individual, school-level and area-level factors affecting variation in SEN provision across different phenotypes (WP2). WP3 assesses the impact of SEN provision on health and education outcomes for specific health phenotypes using a range of causal inference methods to account for confounding factors and possible selection bias. In WP4 we review local policies and synthesise findings from surveys, interviews and focus groups of service users and providers to understand factors associated with variation in and experiences of identification, assessment and provision for SEN. Triangulation of findings on outcomes, variation and impact of SEN provision for different health phenotypes in ECHILD, with experiences of SEN provision will inform interpretation of findings for policy, practice and families and methods for future evaluation.

**Ethics and dissemination** Research ethics committees have approved the use of the ECHILD database and, separately, the survey, interviews and focus groups of young people, parents and service providers. These stakeholders will contribute to the design, interpretation and communication of findings.

## STRENGTHS AND LIMITATIONS OF THIS STUDY

⇒ Education and Child Health Insights from Linked Data (ECHILD) database comprises longitudinal histories of all hospital contacts funded by the National Health Service (NHS) and state-funded schooling for 14.7 million children in England, enabling exploration of outcomes for different health phenotypes over time, and by geographical area and sociodemographic characteristics.

⇒ We define phenotypes in health data, which are recorded independently from processes in schools that lead to special educational needs (SEN) provision and apply different biostatistical and econometric methods to address potential confounding and selection bias.

⇒ We use lived experience evidence from service users and providers to understand varying processes for identifying children who need and are provided with interventions for SEN together with evidence from analyses of ECHILD to strengthen the robustness of our findings and interpretation.

⇒ The provision of SEN is based on organisational and social factors, as well as additional learning needs of children that are not objectively measured in health or education data before intervention.

⇒ The ECHILD database does not capture NHS healthcare outside acute hospital settings, education support at home or in the non-state funded sector, or information on what (if any) SEN provision was received and whether it was appropriate.

## INTRODUCTION

National policies in the UK and in many high-income countries require schools to make adaptations to meet the needs of children who have health, learning or behavioural problems, which impact their ability to learn. These children are

referred to collectively as having special educational needs (SEN). Interventions and adjustments in schools for children with SEN are referred to as SEN provision, and are intended to improve inclusion and participation in education and support children's health and well-being (see online supplemental appendix 1 for details of SEN provision in England).[1] Since 2015, approximately one in six children in England are recorded as receiving any SEN provision each year (see figure 1),[2] and one-third of all children have a record of any SEN provision at least once during their time in education.[3][4]

SEN provision across England is widely regarded as inequitable.[2][4–6] The proportion of pupils with SEN support (a more common type of provision, arranged and funded by the schools, see online supplemental appendix 1) ranged from 7.3% to 17.1% across local authorities in 2018/2019. The proportion with Education, Health and Care Plans (EHCPs, which involve additional, more intensive and higher cost provision for children whose needs cannot be fully met by SEN support, arranged and partly funded by local authorities[2][7]) ranged from 0.8% to 5.0%.[8] Allocation of SEN provision is associated with a variety of factors. According to a recent report, a key factor determining SEN provision is the school, particularly school's previous rates of SEN provision, academy status and previous school inspection outcomes.[4] Other factors include the proportion of academised primary schools and rates of pupils eligible for free school meals at local authority level and pupil-level factors such as attainment at school entry (age 5), ethnic group, child's first language and contacts with social care.[4–6] The annual proportion of children with recorded SEN provision has declined over time, from 20% in school year ending in August 2010 to 14% in 2016. This change seems partly related to the Children and Families Act in 2014 and Special Educational Needs and Disability Code of Practice implemented in 2015, and to reduced funding to local authorities from 2010 (figure 1).[2][9]

Compared with their peers, children with recorded SEN provision experience higher rates of chronic physical and mental health conditions and hospitalisations, and have lower self-reported well-being.[10–12] Recent evidence reviews found that classroom-based SEN interventions improved children's social, emotional well-being and reduced challenging behaviour, and contributed to better mental health outcomes.[13][14] For children with attention deficit hyperactivity disorder, systematic reviews of randomised or quasi randomised controlled trials (RCTs) of interventions similar to SEN provision found improvements in behaviour.[10][15–17] However, there is a lack of RCTs or representative observational comparative studies of the impact of SEN provision, as delivered in routine practice, on health outcomes for a range of health conditions.

Robust evidence that SEN provision improves educational outcomes for pupils with SEN is also scarce.[13] There is moderate evidence that SEN provision in primary schools improves literacy difficulties, socioemotional development and language and communication.[14] A recent evidence review found a weak but positive impact of inclusive education involving additional support for those with additional

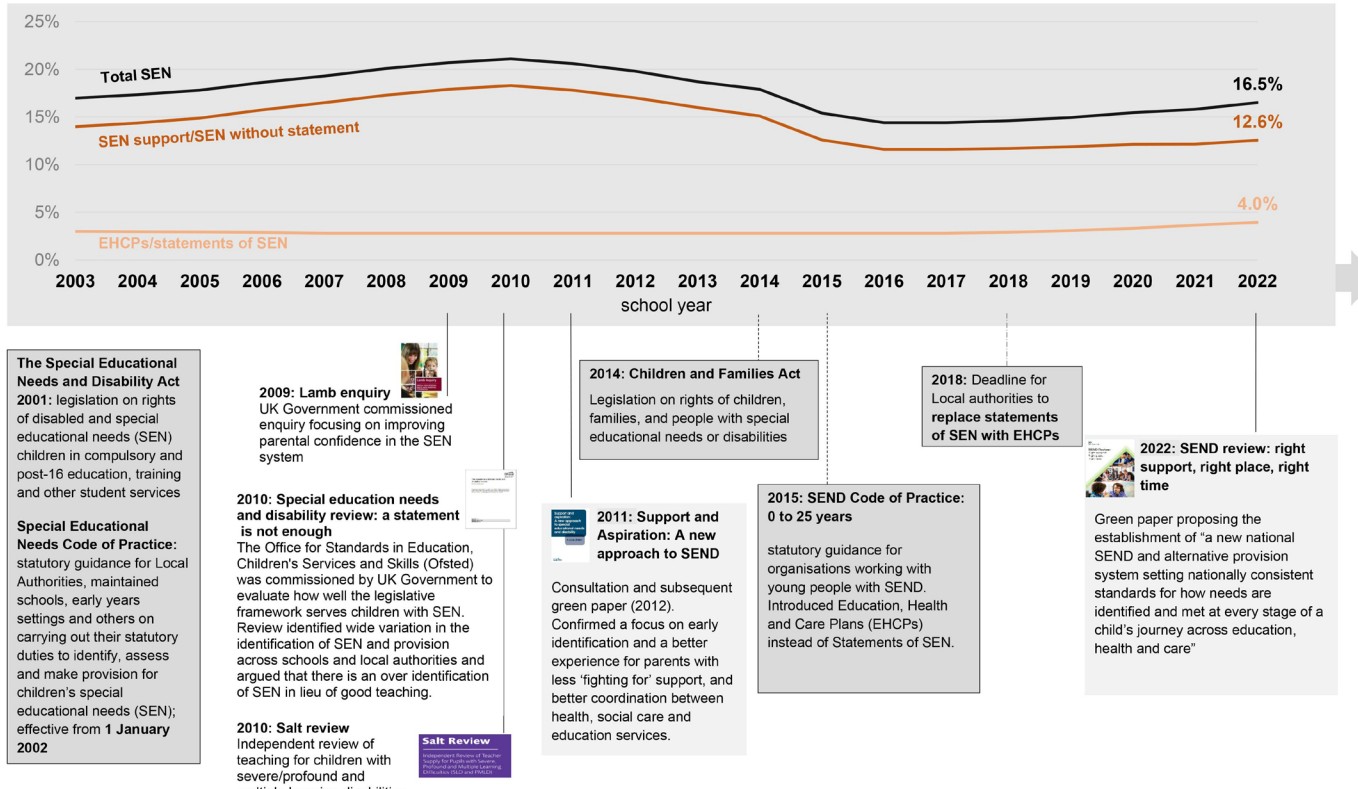

**Figure 1** Percentage of total children with recorded SEN provision per school year, January 2003–2022 (based on DfE statistics).[8][49][50] DfE, Department for Education; SEND, special educational needs or disability.

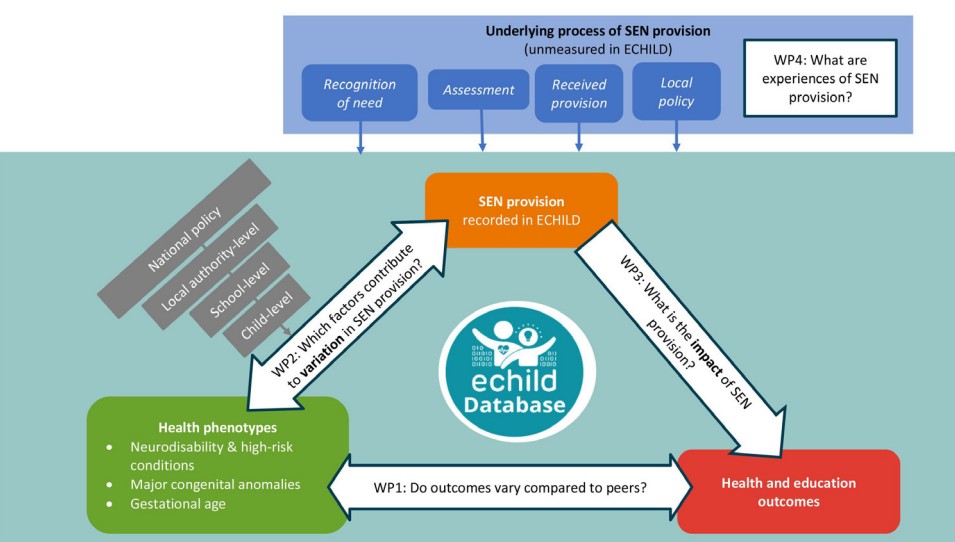

**Figure 2** Conceptual framework bringing together research questions to be addressed by component studies of the HOPE research programme. ECHILD, Education and Child Health Insights from Linked Data; EHCP, Education, Health and Care Plan; HOPE, Health Outcomes of young People in Education; SEN, special educational needs; WP, work package.

learning needs on academic outcomes among pupils without SEN provision.[13] Evidence from population-based observational studies suggests that SEN provision is associated with fewer absences and exclusions among children and young people with neurodisability or mental health conditions.[18]

Given the large proportion of children with SEN provision indicated in school records, the high costs of SEN provision, and static school funding per pupil since 2010,[9] evidence is needed to quantify how SEN provision varies across England and to guide effective intervention to groups of children who are most likely to benefit. The Health Outcomes of young People in Education (HOPE) research programme aims to address these gaps through novel proof-of-concept analyses of the Education and Child Health Insights from Linked Data (ECHILD) database, which links data from all state schools, and contacts with National Health Service (NHS) hospitals in England, and mixed methods (surveys, interviews and focus groups with families and service providers and document analyses) to understand experiences of service users and providers. We will assess two central research questions: (1) which factors contribute to variation in SEN provision in England and (2) what is the impact of SEN provision on health and education outcomes? We will address these research questions for a range of health conditions associated with increased need for SEN provision, which we refer to as health phenotypes.

This umbrella protocol sets out the research plan for the HOPE programme to address these two core questions. We describe four parallel work packages (WP), and in brief, the component studies contributing to each WP. The conceptual framework and proposed research questions are illustrated in figure 2. Separate study protocols for each component study will be preregistered on National Institute for Health and Care Research (NIHR) Open Research platform (https://openresearch.nihr.ac.uk/) and follow relevant reporting guidelines from the EQUATOR

Network (https://www.equator-network.org/, eg, analyses using ECHILD will be reported using REporting of studies Conducted using Observational Routinely-collected Data (RECORD) Reporting Guidelines for studies using linked administrative data).[19]

In the first WP (WP1), we define a range of health phenotypes, that is, health conditions captured in hospital records that are expected to need SEN provision in primary school. We explore how health and education outcomes vary for children with different health phenotypes and compared with unaffected peers. In WP2, we describe how child, social and area-level factors affect variation in SEN provision within phenotypes. In WP3, we apply a range of causal inference methods to address confounding factors (informed by WP2) and possible selection bias to assess the impact of SEN provision on outcomes for children with selected health phenotypes (defined in WP1), also considering timing, duration and level of provision. The ECHILD database contains termly records indicating provision for SEN, but no information about whether or when any provision was actually received, its type or quality. WP4 applies mixed methods to understand geographical variation in local policies and the underlying processes of identification, assessment and provision, and how these processes are experienced by families. Triangulation of findings on outcomes (WP1), variation in (WP2) and impact of (WP3) SEN provision for children with different health phenotypes from ECHILD, with findings on local policies and experiences of SEN provision (WP4) will inform findings for policy, practice and families and methods for future evaluation.

The HOPE research programme started in August 2021 and is expected to end in April 2025. The research programme is ongoing and elements of the programme that have already been completed at the time of publication of this protocol are highlighted in the methods and analyses below.

**Figure 3 table — Coverage of ECHILD datasets, by academic year with refreshes**

Academic year (1st September to 31st August): 1996/97, 1997/98, 1998/99, 1999/00, 2000/01, 2001/02, 2002/03, 2003/04, 2004/05, 2005/06, 2006/07, 2007/08, 2008/09, 2009/10, 2010/11, 2011/12, 2012/13, 2013/14, 2014/15, 2015/16, 2016/17, 2017/18, 2018/19, 2019/20, 2020/21, 2021/22

| Dataset name | Coverage notes |
| --- | --- |
| HES Admitted Patient Care | marker (a) at 1996/97 |
| HES Critical Care | marker (a) at 2005/06 |
| HES Accident and Emergency | marker (b) at 2006/07; marker (a) at 2020/21 |
| HES Emergency Care Data Set (ECDS) | marker (a) at 2018/19 |
| HES Outpatients | marker (c) at 2002/03 |
| HES-ONS Linked Mortality Data | marker (d) at 1996/97 |
| NPD School Census Pupil Level | |
| NPD Pupil Referral Unit Census | marker (e) at 2012/13 |
| NPD Alternative Provision Census | marker (f) at 2012/13 |
| NPD Early Years Census | |
| NPD Absences | marker (g) at 2020/21 |
| NPD Exclusions | |
| NPD Early Years Foundation Stage Profile | marker (h) at 2002/03; markers (g) (g) (i) at 2019/20–2021/22 |
| NPD Key Stage 1 | markers (g) (g) at 2019/20–2020/21 |
| NPD Key Stage 2 | markers (g) (g) at 2019/20–2020/21 |
| NPD Key Stage 3 | marker (j) at 2012/13 |
| NPD Key Stage 4 | markers (k) (k) at 2019/20–2020/21 |
| NPD Key Stage 5 | markers (k) (k) at 2019/20–2020/21 |
| NPD National Client Caseload Information | |
| NPD Children Looked After Return (CLA) | marker (a) at 2005/06 |
| NPD Child in Need Census (CiN) | marker (a) at 2007/08 |

**Figure 3** Coverage of ECHILD datasets, by academic year with refreshes. (a) Partial coverage of an academic year as NPD social care data and HES data are collated by financial year (1 April to 31 March). (b) Partial coverage as HES Accident and Emergency data was experimental and did not have full national coverage. (c) Partial coverage as HES outpatient data was experimental and did not have full national coverage. (d) Partial coverage of an academic year as ONS mortality data was first linked to HES in January 1998. (e) The Pupil Referral Unit Census was subsumed in the School Census Pupil Level from 2013/2014. (f) The Early Years Census included 3 years and 4 years between 2007/2008 and 2012/2013. From 2013 to 2014, it includes 2–4 years. (g) Not collected to help reduce the burden on educational and care settings during the COVID-19 pandemic. (h) Partial coverage as between the 2002/2003 and 2005/2006 academic years, data only on a 10% sample of children. (i) To be included, but not available yet. (j) Key stage 3 assessments ceased after 2012/2013. (k) Data not provided with standard institutional identifiers in 2019/2020–2020/2021, as evaluation of individual institutional performance is not permitted. CiN, Child in Need Census; CLA, Children Looked After Return; ECHILD, Education and Health Insights from Linked Data; HES, Hospital Episode Statistics; NPD, National Pupil Database, ONS, Office for National Statistics.

## METHODS AND ANALYSIS

### Data sources and study measures for WP1–3

#### ECHILD database

The ECHILD database links routinely collected administrative data on health and education in England. Currently, ECHILD includes all children and young people aged 0–24 years in England who were born between 1 September 1995 and 31 August 2021 (approximately 14.7 million individuals).[20] Health and education datasets were linked by NHS England using a multistep deterministic linkage algorithm, described in detail elsewhere.[21] Linkage rates were high and increased over time (92% of school pupils born in academic year 1990/1991 were linked to a hospital record, compared with 99% of pupils born in 2004/2005).[21]

Health data consist of Hospital Episode Statistics (HES), a national database that includes dated information on all NHS acute hospital care and mortality data (see figure 3 for details of data coverage by academic year).[22] Nearly all children born in England are born in NHS hospitals (97%) but HES excludes births in private hospitals or at home. Children can be followed from their birth admission through all subsequent NHS hospital contacts.[23 24]

Education records are collated in the National Pupil Database and include information on children's registrations in schools, attainment scores at ages 5, 7 11, 16 and 18 (see Study Glossary in online supplemental appendix 2 for details), and number of half-day absences and exclusions in each 13-week term. SEN provision is recorded each term (annually prior to 2005/2006) for all children in state-funded education (93% of all children) from the academic year starting in September 2001 onwards (see figure 3).[25] Education data capture the category of recorded SEN provision (SEN support or EHCP) and main reason for SEN provision. Since 2014/2015, reasons for SEN include language or communication, moderate or severe learning disability, autism, sensory impairment, physical disability, or social, emotional or mental health needs (see online supplemental appendix 1 for details). We are in the process of enhancing ECHILD with school characteristics, such as type of school (eg, mainstream or special) or teacher-pupil ratio, from a range of publicly available data.[26–28]

All analyses of ECHILD across WP1–3 will use shared definitions of study population, SEN provision and outcomes, as described in more detail below. The analyses of the ECHILD database will be a proof of concept, restricted to children attending primary school and three groups of health phenotypes (described in detail below). Analyses of all age groups, and all possible health phenotypes, types of SEN provision, and possible comparisons are beyond the scope of the HOPE programme, but can be informed by our methods. We will publish our methods and code to enable others to reproduce and extend our analyses using ECHILD.

### Study cohort and health phenotypes

Our target population is primary school-aged children (enrolled in school in year 1 aged 5/6 and followed to year 6 aged 10/11), who were born in an NHS hospital in England and had a birth admission recorded in HES data from September 2002 onwards. We follow up all children from birth and use information about risk factors at birth (such as gestational age, birth weight) and health phenotypes recorded in health data before the start of primary school.[23 24] We can then evaluate exposure to SEN provision at the start of primary school and the impact on subsequent health and education outcomes. We analyse three groups of health phenotypes as we hypothesise that the impact of SEN provision on health and education outcomes will vary for children with different health phenotypes.

Not all children in English primary schools have a birth record in HES. Between 1 September 2009 and 31 August 2017, there were 5 004 354 children entering primary school (recorded in school census in year 1, aged 5–6 years old), of whom 94% had a linked HES record and 80% had a linked birth record (the linkage rate increased over time, see (online supplemental table 1 in appendix 3). As these numbers are large, we can focus on specific or even uncommon phenotypes.

We will use clusters of coded clinical information in the ECHILD database to define health phenotypes that represent health conditions associated with learning impairment or need for additional educational support. We have defined three groups of health phenotypes which capture populations with different levels of need for SEN provision:

### Neurodisability and other high-risk conditions

The first group of health phenotypes comprises neurological conditions or complex systemic health problems reported to be associated with learning impairment or behaviours that require SEN provision. These include neurodisability such as autism or learning disabilities,[29 30] cerebral palsy[31] or epilepsy.[32] The list of health phenotypes has been derived from an overview of systematic reviews and population-based observational studies (see online supplemental appendix 3 for overview of search terms) and discussions with clinical experts and service providers. We developed coding algorithms for these health phenotypes based on combinations of diagnostic and procedure codes, where possible from previously validated code lists. As part of validation, we will compare the cumulative incidence and mortality rates by age for each health phenotype with external population studies (eg, from national surveys and disease registries) and changes over time to assess consistency of recorded diagnoses in hospital records and to further refine the phenotyping algorithm. A detailed phenotyping paper is in preparation.

Preliminary findings for children with neurodisability or high-risk conditions include 50 high-risk health phenotypes recorded in hospital records before the age of 5. These account for approximately 5% of all children starting primary school in 2008/2009–2018/2019, 10% of children with any recorded SEN provision during primary school and 30% of those with an EHCP. Some of the included conditions (such as autism or learning disability) are likely to be under-reported in hospital records.[33]

### Major congenital anomalies

The second group includes children with major congenital anomalies (MCAs), as children with MCAs are likely to require support from specialist services and have a diagnosis recorded in hospital admissions records, creating a reliable phenotype defined before entry to primary school among children whose need for SEN provision is likely to vary.[34] We are using a code list of International Classification of Diseases version 10 (ICD-10) diagnoses developed by EUROCAT—a European Congenital Anomaly Registry (https://eu-rd-platform.jrc.ec.europa.eu/eurocat),[35 36] which groups MCAs into 12 body system groups and includes 25 specific subgroups.

Our preliminary findings suggest that children with MCAs recorded in the first year of life account for 3.5% of the primary school population in 2008/2009–2018/2019, 5.5% of children with any recorded SEN provision during primary school, and 13.6% of those receiving EHCPs in mainstream school or attending a specialist school.

### Whole population phenotype: gestational age

Finally, we are using week of gestational age at birth to assess the gradient in impact of SEN provision across the whole population of children, stratified according to different levels of underlying need. This approach is supported by the finding that each week of birth before 40 weeks of gestation is associated with reduced school attainment scores and an increased risk of SEN intervention.[11 37 38] Approximately 4.5% of children in primary school in 2008/2009–2018/2019 were born preterm (at <37 weeks' gestation), accounting for 6.0% of children with any recorded SEN provision during primary school, and 8.4% of those receiving EHCPs.

### Health and education outcomes

We focus on outcomes that can be measured in hospital and education data: unplanned (accident and emergency and unplanned admissions) and planned hospital contacts (planned or elective admissions and outpatient appointments), school attainment (as proxy measure for cognitive function), and rates of school absences (see box 1 for study measure definitions).

Follow-up of outcomes will cease before the onset of the COVID-19 pandemic. COVID-19 had a significant impact on the well-being of young people.[39] Lockdowns in England affected children's access to school and the frequency of hospital contacts captured in ECHILD data. Planned and unplanned admissions, and outpatient appointments reduced substantially during the COVID-19 pandemic, with the largest reductions in children with indicators of vulnerability (such as preterm birth, a chronic condition, recorded SEN or social care record).[12 40] School attainment measures were not collected during the pandemic to help reduce the burden on educational and care settings. In the HOPE

## Box 1 Key study measures derived from ECHILD database

**Measures derived from health data:**

Accident and emergency (A&E) department contact rate: Defined as the number of days with at least one A&E contact, divided by person-time at risk during the study period (eg, time from start of year 1 until the end of year 6 or death).

Unplanned/planned admission rate: Defined as the number of unplanned or planned hospital admissions in NHS-funded hospitals in England, divided by person-time at risk during the study period. Admissions will be classified as planned/unplanned according to the admission method recorded in the first episode of care. Consecutive admissions with readmission within 1 day of discharge (eg, hospital transfers) will be treated as part of the same admission. Time spent in hospital during an admission will be taken out of the person-time at risk as once a child is in hospital they cannot be at risk of a new admission.

Outpatient department (OPD) appointments and attendances: Defined as the number of days with at least one OPD contact, divided by person-time at risk during the study period (eg, time from start of year 1 until the end of year 6 or death).

**Measures derived from education data:**

Absence rate: Schools are required to take attendance registers twice a day, for morning and afternoon sessions. Absence rates are defined in line with the definition used by the Department for Education as the total number of absent sessions (including authorised and unauthorised absences) divided by the total number of possible sessions during the study period.

Standardised attainment measures: We derive standardised attainment measures using recorded scores from national tests in reading, writing and maths at the end of year 2 (aged 7, key stage 1) and at the end of year 6 (aged 11, key stage 2, see online supplemental appendix 2 for details of key stages). Standardised test scores are calculated using mean and standard deviation (SD) of the test scores of all pupils in a given academic year. We will present the proportion of children not assessed (ie, who did not have an assessment record) and average score for those assessed by study population.

School readiness indicators: We use scores from teacher assessments of children's development across multiple areas of learning, carried out in the final term before year 1 (Early Year Foundation Stage Profile, EYFSP, see online supplemental appendix 2 for details). Standardised EYFSP scores are calculated using mean and SD of the EYFSP scores of all pupils in a given academic year. We will present the percentage of children who were not assessed (ie, did not have an assessment record), the proportion of children not reaching Good Level of Development (defined by Department for Education using a subset of EYFSP scores) and average scores for those assessed by study population.

SEN provision: We use four categories (which may be merged for some analyses) in the following descending hierarchy for a specified time period (eg, year 1):

1. Enrolment in specialist provision (including special school or alternative provision).
2. EHCP (including 'Statement of SEN' or 'EHCP') in mainstream school.
3. SEN support (including 'School Action', 'School Action Plus' or 'SEN support') in mainstream school.
4. No SEN provision.

ECHILD, Education and Child Health Insights from Linked Data; NHS, National Health Service; SEN, special educational need.

programme, we will, therefore, limit our analyses to outcomes recorded before the start of COVID-19 pandemic.

### Recorded SEN provision

Schools record information on children identified as needing SEN provision (SEN support or EHCP) in school censuses returned to the Department for Education (DfE). We refer to this recording as SEN provision throughout the protocol, although we acknowledge that an indication of SEN provision in educational records does not evidence that SEN provision is actually received or whether it is appropriate, as these data are not recorded by schools.[25]

We categorise SEN provision in a descending hierarchy for a specified time period (eg, during school year 1 or across all of primary school; details shown in box 1), separating any enrolment in a special school or alternative provision (where the vast majority of children have recorded SEN provision), an EHCP in mainstream school, SEN support in mainstream school, and no recorded SEN provision. These categories have been selected due to differences in the presumed type of provision, in the criteria for provision (eg, formal assessment is required for EHCP but not for SEN support) and substantial differences in associated costs.[2]

SEN provision changed following government education reforms in 2014/2015, when EHCPs gradually replaced Statements of SEN, and SEN support replaced SEN without Statement (which grouped two levels of provision: School Action and School Action Plus). We will report changes in recorded SEN provision over time and address potential impacts of these changes in the design of analyses.

### Data sources for WP4
#### National online survey

Research for WP4 to date includes three online surveys aimed at (1) children and young people, (2) parents/carers and (3) service providers (health, education and local authority professionals). The surveys document variation in local experiences of identification and assessment of need, and provision of SEN intervention. Detailed information on survey design and findings will be published separately. In brief, the surveys were developed through a scoping review to identify previous questionnaires and co-designed with stakeholder groups of young people, parents/carers and professionals working in education or health with children who have SEN. Each survey underwent three rounds of extensive piloting with the respective advisory groups from the HOPE study. This helped to: (1) ensure that the questions and response options matched the lived experience of participants, (2) test accessibility and usability for respective participants and (3) ensure no technical difficulties existed when completing the survey across multiple device types (eg, smartphone, laptop). Data were collected using REDCap and the surveys were disseminated via social media (Twitter, Instagram and Facebook), and through professional networks (GOV.UK Notify service, Parent and Carer forums, and stakeholder group contacts). These networks were used to maximise

the recruitment of all three groups from each of the nine administrative health regions in England.

In total 1714 participants took part from across England including: 77 young people aged 11–27 years, 772 parents and carers, and 865 service providers (those working in/ closely with education settings, the health services, local authorities). Short summaries of the key findings at the time of submission of this protocol from initial analyses are published on the study website (https://dev.psychiatry.cam.ac.uk/hope-study-health-outcomes-for-young-people-throughout-education/) and more detailed papers are in preparation.

## Qualitative studies of children, young people, parent/carer and practitioner experience

We are conducting two qualitative studies (including interviews and focus groups) to explore the experiences and beliefs of children, young people, parents/carers and practitioners about SEN provision to assist the interpretation of our quantitative analyses using ECHILD. Both qualitative studies recruited from the survey respondents who agreed to recontact in the national survey described above.

### Children, young people and parents/carers

We conducted semistructured interviews with children and young people (supported by carer if they wished) and separately with parents/carers. The topic guide covered the identification, assessment and provision of support for their or their child's SENs using a time-line approach. Sixteen interviews with children and young people and 22 with parents/carers were completed between May and June 2023 and will be analysed using a framework approach (see the Analysis plan section for details).[41]

### Practitioner working in health and education

We will conduct three focus groups of up to 10 practitioners on each of the major areas identified by our stakeholder groups (identification, assessment and provision of SEN intervention), providing a total of 9 focus groups involving up to 90 practitioners working in relation to SEN across health, education, local authority or social services. Each area of focus will have its own topic guide and each series of focus groups will be facilitated by a team of two researchers and a parent observer. The latter will provide their opinion of the discussion in focused debriefing sessions after each focus group. Analysis will follow a framework approach,[41] and parent observers will contribute to the interpretation of our findings at a final joint meeting.

## Analysis plan
### WP 1: describing health and education outcomes

We will carry out separate descriptive studies for each of the three groups of health phenotypes. We will estimate rates of planned and unplanned hospital contacts and educational outcomes during primary school for children with and without each health phenotype. We will use appropriate generalised linear models for each outcome (eg, Poisson regression for rates, logistic regression for binary outcomes). Findings from these analyses will indicate whether there are differences in outcomes across subgroups within each health phenotype and compared with their peers, to inform analyses in WP3.

### WP 2: variation in recorded SEN provision

We will use ECHILD to understand variation in SEN provision for children with different phenotypes. We will examine how proportions of children with recorded SEN provision changed over time for children with different health phenotypes (eg, MCAs), and the percentage in their variation that is explained by factors at the individual-level, school-level and local authority-level using appropriate multilevel regression models. These analyses will determine whether the impact of SEN provision (examined in WP3) can be evaluated using natural policy experiment designs (eg, due to changes in policy over time) and instrumental variable analysis. These analyses will also generate findings on variation in SEN provision according to child level demographic, social and service use characteristics.

### WP 3: impact of SEN provision on outcomes

We will use a range of biostatistical and econometric methods to explore the impact of SEN provision on health and educational outcomes across selected health phenotypes, and triangulate findings from analyses using different methods (briefly described below). First, we will examine the impact of recorded SEN provision at a given point in time, in the first year of compulsory education (year 1). Second, we will assess the impact of the duration of SEN provision, appropriately controlling for likely time-varying factors that may be affected by SEN provision (informed by findings from WP2) and additionally influence future SEN provision. We anticipate separate studies focussing on specific health phenotypes that represent conditions that are relatively similar in their need for SEN and well characterised in health data. For example, cleft lip and palate has been selected from the MCA phenotypic group as it is reliably denoted by diagnostic and operation codes. We will select a well-defined phenotype within the neurodisability group and we also plan to compare exposure to SEN provision for children within defined strata of gestational age at birth across the whole population.

For all causal analyses, we will use the target trial emulation (TTE) framework to guide the creation of study cohorts that correspond to the specific phenotypes and exposure levels (categories of SEN provision) of interest.[42 43] TTE consists of first designing an ideal pragmatic trial that would address the question of interest, and then emulating it as closely as possible using observational data. The advantage of this approach is the avoidance of biases in the design stage, for example, immortal time bias and prevalent-case bias, which have affected real-world data studies in the past.[42] Directed acyclic graphs will be used to draw our assumptions about the causal structures influencing what we are studying and to identify relevant confounding variables.[44] An outline of the components of our causal investigations is given in online supplemental table 2 in appendix 3 where these steps are illustrated

using the exemplar of one MCA phenotypes: children born with cleft lip and palate. The study protocol for this study has been registered as a preprint.[45]

As ECHILD comprises observational data, a major challenge for each of our causal investigations will be how to address the bias introduced by confounding by indication, as well as possible selection bias due to incomplete linkage across health and education databases. We will contrast estimates of the impact of SEN provision on health and educational outcomes using a variety of complementary methods that rely on measuring all of the confounders, such as regression adjustment, g-computation, inverse probability weighting of marginal structural models (with different approaches to modelling the propensity score), or econometric methods that try to deal with unmeasured confounding by exploiting natural experiments (such as differences-in-differences or interrupted time series), or instrumental variables.

### WP 4: understanding policy variation and lived experiences of SEN provision

We will conduct an overarching synthesis of the findings from the national survey, parent and child/young person interviews and focus groups with practitioners, in relation to the quantitative findings from WP1 to WP3. The planning of this work draws from previous similar triangulation of quantitative and qualitative systematic reviews.[17] We will first create a matrix to demonstrate where these different data sources provide similar or conflicting signals. We will then work inductively from the surveys, focus group and interview findings to generate hypotheses about contextual elements that may influence the recording of data in ECHILD, or health outcomes of SEN provision. This approach draws from 'complexity theory' which assumes that any psychosocial intervention must be considered in terms of the larger environment in which it is located.[46] We will also work deductively from the results of analyses in WP1-3 to explore potential relationships between SEN provision and health outcomes, as well as exploring what WP4 findings suggest about factors associated with health outcomes in SEN provision. The aim of this approach is to clarify potential explanations for the findings of WP1-3 and to inform future work in this area. Analysis under the two approaches described above will proceed iteratively and in parallel.

To understand local variation in SEN provision we reviewed publicly available documents on the support available for local children with SEN, referred to as 'local offer'. We are assessing the quality of available information against 51 criteria outlined within the SEND Code of Practice to determine to what extent local authorities in England are providing clear, comprehensive, accessible and up-to-date information about available SEN provision and how to access it. Second, we are examining reports from all of the local area SEND inspections published by the Office for Standards in Education, Children's Services and Skills over the past 7 years to examine how effectively local authorities fulfil responsibilities for children and young people with SEN. By collating these documents and assessing their commonalities and differences, we aim to gain an understanding of variation in good and bad practice in SEN provision.

### Patient and public involvement

The HOPE study was developed in response to consultations about the need for the ECHILD database with parents and charities supporting children with chronic health conditions and their families.[47 48] We have established three stakeholder groups: young people, parents/carers and professionals working in education or health with children who have SEN. We are collaborating with staff in schools to enable young people with additional learning needs or disabilities to contribute to these advisory groups. In addition, we are iteratively presenting our study plans and preliminary results to parents/carers and young people through research advisory groups at UCL and University of Exeter. These consultations are contributing evidence to all four WPs and dissemination of the research. The HOPE Study Steering Committee includes two parents of children with disability and will adhere to NIHR requirements for payment for time and expenses of lay contributors.

### ETHICS

Existing research ethics approval has been granted for analyses of the ECHILD database for the purposes set out in the HOPE study (20/EE/0180). Data access is also controlled by agreements with NHS Digital and the DfE. The data contain no identifiers or sensitive dates and data can only be used within the Office for National Statistics Secure Research Environment by approved researchers, with strict statistical disclosure controls of all outputs of analyses (eg, tables or figures). Details are published here in our privacy notice (https://www.ucl.ac.uk/child-health/research/population-policy-and-practice-research-and-teaching-department/cenb-clinical-4#).

Separate ethics approval has been approved for the mixed-methods research (national survey, interviews and focus groups) involving service users (young people and parents) and service providers (PRE:2021.058). Parents consented for their own involvement and also for their child if under the age of 16. Young people aged 16 or over consented and younger children were asked for assent to their participation using a similar process.

### Dissemination

We will present preliminary findings to diverse audiences (academics, analysts at DfE and Department of Health and Social Care, and our stakeholder groups as well as other groups of service users and providers) through seminars, question and answer sessions, workshops and consultations during the study. We will incorporate feedback into final outputs, which will include peer-reviewed journal articles, the final study report to funder, and short briefing reports and infographics for non-academics published on the study website.

We will publish our methods and code to enable others to reproduce and extend our analyses using ECHILD. ECHILD can be accessed by accredited researchers through application via the ECHILD team (www.ucl.ac.uk/child-health/echild) and the Research Accreditation Panel. Meta-data and code relating to the HOPE study will be signposted on the study website and made available in the ONS secure environment and in code repository (including on ECHILD GitHub page: https://github.com/UCL-ECHILD). We will hold workshops to promote wider use of findings from the HOPE study for causal analyses of education interventions on health. Examples from the HOPE study will be incorporated into short courses on causal methods and on how to use the ECHILD database.

The HOPE study aims to build the evidence base for fairer and more effective SEN provision and, by informing national and local policy and the public and changing practice, to improve health and education outcomes of children with SEN.

**Author affiliations**
[1]UCL Great Ormond Street Institute of Child Health, UCL, London, UK
[2]Department of Psychiatry, University of Cambridge, Cambridge, UK
[3]Department of Health and Community Sciences, University of Exeter Medical School, Exeter, UK
[4]Institute of Psychiatry, Psychology and Neuroscience, King's College London, London, UK
[5]South London and Maudsley NHS Foundation Trust, London, UK
[6]The Peninsula Childhood Disability Research Unit, University of Exeter Medical School, Exeter, UK
[7]UCL Institute of Ophthalmology, UCL, London, UK
[8]Faculty of Education, University of Cambridge, Cambridge, UK
[9]UCL Social Research Institute, UCL, London, UK

**Acknowledgements** We would like to acknowledge the contribution of members of the HOPE study steering committee. We gratefully acknowledge all children and families whose deidentified data are used in this research. We are grateful to the Office for National Statistics (ONS) for providing the trusted research environment for the ECHILD Database. ONS agrees that the figures and descriptions of results in the attached document may be published. This does not imply ONS' acceptance of the validity of the methods used to obtain these figures, or of any analysis of the results. The ECHILD Database uses data from the Department for We would like to acknowledge the contribution of members of the HOPE study steering committee. We gratefully acknowledge all children and families whose de-identified data are used in this research. We are grateful to the Office for National Statistics (ONS) for providing the trusted research environment for the ECHILD Database. ONS agrees that the figures and descriptions of results in the attached document may be published. This does not imply ONS' acceptance of the validity of the methods used to obtain these figures, or of any analysis of the results. The ECHILD Database uses data from the Department for Education (DfE). The DfE does not accept responsibility for any inferences or conclusions derived by the authors. This work also uses data provided by patients and collected by the National Health Service as part of their care and support. Source data can also be accessed by researchers by applying to NHS England. This work was produced using statistical data accessed via the ONS Secure Research Service. The use of this data in this work does not imply the endorsement of the ONS in relation to the interpretation or analysis of the statistical data. This work uses research datasets which may not exactly reproduce National Statistics aggregates. The views in this publication do not necessarily reflect the views of UCL.

**Contributors** RG is the principal investigator of the HOPE research programme and KH, TF, LD and BDS lead each of the four work packages, KB-H, JD, SL and JR are coinvestigators of the programme. AZ and RG produced the first draft of this manuscript, AZ, KL and ML created figures for the manuscript. AZ, KL, VN, AS and ML are responsible for preparing ECHILD data for analyses. JM, IW, SB, JS, KB and TF are responsible for conceptualisation and implementation of mixed methods studies. BDS, LD, KL, VN and LK are responsible for design and implementation

of quantitative causal inference analyses using ECHILD database. AZ, KH, RG and KL are responsible for design and implementation of descriptive analyses using ECHILD database, with clinical input from JD, SL and JR on defining study populations. JS, JM, IW, KB, KB-H, TF, RG, KH, AZ and VN were involved in public engagement activities to inform the study. All authors edited the manuscript. All authors contributed to and are responsible for the final design of the study. All authors read and approved the final manuscript.

**Funding** This study/project is funded by the National Institute for Health Research (NIHR) under its Programme Grants for Applied Research Programme (Grant Reference number NIHR202025). The views expressed are those of the author(s) and not necessarily those of the NIHR or the Department of Health and Social Care. ECHILD is supported by ADR UK (Administrative Data Research UK), an Economic and Social Research Council (part of UK Research and Innovation) programme (ES/V000977/1, ES/X003663/1, ES/X000427/1). RG was supported by Health Data Research UK (grant No. LOND1), which is funded by the UK Medical Research Council and eight other funders and by a NIHR senior investigator award. Research at UCL GOS ICH is supported in part by the NIHR Great Ormond Street Hospital Biomedical Research Centre. The funders had no role in study design, data collection and analysis, decision to publish, or preparation of the manuscript.

**Disclaimer** The views expressed are those of the author(s) and not necessarily those of the NIHR or the Department of Health and Social Care. The funders had no role in study design, data collection and analysis, decision to publish, or preparation of the manuscript.

**Competing interests** None declared.

**Patient and public involvement** Patients and/or the public were involved in the design, or conduct, or reporting, or dissemination plans of this research. Refer to the Methods section for further details.

**Patient consent for publication** Not applicable.

**Provenance and peer review** Not commissioned; externally peer reviewed.

**ORCID iDs**
Ania Zylbersztejn http://orcid.org/0000-0003-1035-1448
Lucy Karwatowska http://orcid.org/0000-0002-6519-5190
Kate Boddy http://orcid.org/0000-0001-9135-5488
Johnny Downs http://orcid.org/0000-0002-8061-295X
Tamsin Ford http://orcid.org/0000-0001-5295-4904
Katie Harron http://orcid.org/0000-0002-3418-2856

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
