## [Reviewer comments · BMJ Open]

ARTICLE DETAILS

TITLE (PROVISIONAL)	Evaluation of variation in special educational needs provision and its impact on health and education using administrative records for England: umbrella protocol for a mixed methods research programme
AUTHORS	Zylbersztejn, Ania; Lewis, Kate; Nguyen, Vincent; Matthews, Jacob; Winterburn, Isaac; Karwatowska, Lucy; Barnes, Sarah; Lilliman, Matthew; Saxton, Jennifer; Stone, Antony; Boddy, Kate; Downs, Johnny; Logan, Stuart; Rahi, Jugnoo; Black-Hawkins, Kristine; Dearden, Lorraine; Ford, Tamsin; Harron, Katie; De Stavola, Bianca; Gilbert, Ruth

VERSION 1 – REVIEW

REVIEWER	Christin , Ellermann Universität Potsdam, Faculty of Health Sciences Brandenburg
REVIEW RETURNED	10-Mar-2023

GENERAL COMMENTS	Thank you for the opportunity to review the manuscript. I think your work is really important in addressing the barriers of lack of SEN recording and intervention, and the impact of SEN intervention on health and education outcomes. However, I think that the protocol needs to be revised in terms of how exactly the results are to be collected in the individual studies and how they are to be incorporated into the subsequent work packages. In this respect, common reporting guidelines for quantitative and qualitative studies should be taken into account in order to make the a priori planned process transparent (e.g. PRISMA statement, SRQR statement). Overall, the protocol is not well structured, central research questions and hypotheses for the individual studies as well as approaches are not reported transparently. In particular, the methods should be revised and details of the planned studies should be added. It is unclear which studies have already been carried out, as the project started in August 2021. If the literature search or data analyses has already been completed, the main results should already be mentioned in the protocol. Perhaps the protocol could be rewritten to include only those studies that are still ongoing.
--

REVIEWER	French, Robert Cardiff University School of Medicine, Medicine
	I reviewed the information governance of the ECHILD dataset and study as part of my work on the IGARD panel for NHS Digital (now

	the AGD panel for NHS England). I do not feel this limits my capacity to objectively review the protocol.
REVIEW RETURNED	23-Apr-2023

GENERAL COMMENTS	A very interesting and well-presented protocol, thank you for the opportunity to review this draft. My primary challenge as a reader was that the broad scope of what is planned within the protocol meant that there was often insufficient information to really grasp the detail of what the researchers were actually going to do for some strands. There was good detail on the context and the data, but it felt less clear on what analysis would take place. In places there was a lot of detail, for example for "exploring variation in recorded SEN", but for the "causal workstream" and "Assessing the Impact of recorded SEN on outcomes" it was less clear. This was partly justified, for example, that the researchers would consult with patients on what were the most relevant outcomes Also, that reviews of the literature would help inform casual approaches such as potential IVs. However, it still felt a bit unbalanced and more like a grant proposal than a protocol. My suggestion would be to add more detail for Aps 3-4 in "study Design". In methods, I would consider trying to be more specific wherever possible, particularly in the first paragraph of "Assessing the impact of recorded SEN on outcomes". If more detailed protocols are planned for some of these workstreams it may be worth flagging that here, since that would help justify being less detailed in this protocol? Abstract: "state-funded hospitals" – is this the right term for the data included in HES? Intro: "experiencing deprivation" -> "deprivation levels" ? Intro: "Social care" data pops up without earlier reference to where it comes from, readers may think this is another dataset Data source: "ECHILD follows" -> "ECHILD includes" ? Data source: I may have missed it but it didn't list who did the linkage Data source: Q. does "birth admissions" include all births or only those connected with a hospital admission? Dissemination: If possible, please set up the GitHub repository (if it not done already) and provide the link to the repository here in the protocol Figure 1: This was helpful for context. Figure 2 (labelled as Figure 1!): It might just be me, but I did not find this particularly clear or helpful and wonder if the authors could think of ways to make this more easily understandable. Figure 3: It may be helpful to add a note to say what is planned (if anything) for 2021/22 data Great that you have included a glossary.
---

	Glossary (cf Figure 2): You use "statement" in Figure 2, could that be referenced in your glossary? You may want to add other historical terms like "Action", "Action Plus", to help older readers. Also "Key Stage" Glossary (or elsewhere): Make it clear the 'scale' on which SEN is measured eg lowest level is X, highest level is Y. (I think this must have been covered and I've missed it, but noting just in case)
--	--

VERSION 1 – AUTHOR RESPONSE

Reviewer: 1

Dr. Ellermann Christin , Universität Potsdam Comments to the Author:

Thank you for the opportunity to review the manuscript. I think your work is really important in addressing the barriers of lack of SEN recording and intervention, and the impact of SEN intervention on health and education outcomes. However, I think that the protocol needs to be revised in terms of how exactly the results are to be collected in the individual studies and how they are to be incorporated into the subsequent work packages. In this respect, common reporting guidelines for quantitative and qualitative studies should be taken into account in order to make the a priori planned process transparent (e.g. PRISMA statement, SRQR statement).

Thank you for all your comments. In the last paragraph of the introduction (p6) we explain that this is a protocol for a mixed methods research programme, consisting of multiple component studies. We clarify that detailed protocols for each component study will be reported separately using relevant reporting guidelines from the EQUATOR Network. For example, analyses using ECHILD database will follow the RECORD checklist. In addition, causal analyses will be pre-registered on NIHR Open Research platform (<https://openresearch.nihr.ac.uk/>).

We have not included specific details for all the component studies as there is not sufficient space to transparently report details of several studies and the purpose of the programme is partly to develop and then implement the study protocols. Please see a similar 'umbrella' protocol by Craig et al. <https://bmjopen.bmj.com/content/12/4/e061340>

Revised paragraph on p6 now reads as follows:

“This protocol sets out a mixed methods research programme, with multiple component studies (conceptual framework and proposed research questions are illustrated in Figure 2). The programme integrates quantitative analyses of the Education and Child Health Insights from Linked Data (ECHILD) database (see data resources) with mixed quantitative and qualitative methods to understand variation in identification, assessment and provision for SEN, and how these processes are experienced by families. We will explore which causal inference methods can be used to provide valid evidence of the impact of SEN provision on health and education outcomes, and in which contexts. Each component study in the programme will be reported using relevant reporting guidelines from the EQUATOR Network (<https://www.equator-network.org/>, e.g.: analyses using ECHILD will be reported using RECORD guidelines for studies using linked administrative data).⁴⁵ The programme started in August 2021 and is expected to end in March 2025. We consulted children and young people who chose to name this research programme HOPE: Health Outcomes of young People in Education (HOPE).”

Overall, the protocol is not well structured, central research questions and hypotheses for the individual studies as well as approaches are not reported transparently. In particular, the methods should be revised and details of the planned studies should be added. It is unclear which studies have already been carried out, as the project started in August 2021. If the literature search or data analyses has already been completed, the main results should already be mentioned in the protocol. Perhaps the protocol could be rewritten to include only those studies that are still ongoing.

In the introduction, we now state the two central research questions addressed by the research programme (paragraph 1 on page 6) and our hypothesis about impact of SEN provision on health and educational outcomes (paragraph 2 on page 6).

We have revised the methods section and added more detail to clarify:

- The proposed study population – our analyses will focus on children in primary school who were born in England and examine 3 groups of health phenotypes
- Considered study measures – we provide definitions of health and educational outcomes, and categories of SEN provision, which will be derived from ECHILD and shared across studies
- Analysis plan, providing an overview of planned analyses each component studies (although as noted above, we are not including specific details), including descriptive analyses of outcomes across phenotypes, examining variation in SEN provision, mixed-methods work to understand live experiences of SEN provision, and finally causal analyses to examine impact of SEN provision.

We now also clarify which analyses were carried out and provide provisional results where available. For example, in “Data sources \ National online survey” section we clarify that the national online survey has already been carried out (p7) and provide summary results on number of participants. These data will inform further qualitative components of the study. Under “study population” we provide preliminary findings on number children entering primary school who have a linked birth record in ECHILD (last paragraph, p7), number of children captured by the three health phenotype groups, and proportion within each group with SEN provision (paragraphs 2, 4, 5 on p8). Note that these studies are ongoing and these findings are preliminary.

Please find below a few comments on some of the sections and phrases.

- Line 56: How do school identify SEN support? The school itself? Who at the school is responsible for the assessment?

We have revised the introduction and expanded overview of SEN provision in England (p4-5). We provide more detailed information on assessment for SEN support and EHCP in paragraphs 1-2 on p5 as follows:

SEN support: *“The first assessment for SEN support is usually carried out by the school’s teachers, Special Educational Needs Coordinator (SENCo), or after class teachers, who seek to identify children making less than expected educational progress or with additional social needs relative to their peers.”*

EHCPs: *“An assessment for an EHCP can be requested by parents, schools or health or social care professionals. The assessment is carried out by the local authority, who are required to fill in a legal document setting out the special measures to be provided to meet the child’s needs across education, health and social care.”*

Methods

This is not a methods description. In the methods section, I would expect information on exactly how quantitative and qualitative research is planned:

- how will the systematic search be conducted? (which databases and why, what is the research question, what are the search terms, what kind of studies are searched, what are the inclusion and exclusion criteria, is the whole search and extraction process done in a peer process, etc.).

Our aim for literature review was to obtain a list of conditions associated with higher need for SEN provision, which we can then further discuss with experts and validate in ECHILD database. We now clarify on p8 in section “a) Neurodisabilities and other high-risk conditions” that we conducted a review of systematic reviews and cohort studies as part of an iterative process that includes discussions with clinical or service experts as well as review of results on coding frequency and correlated codes to develop a list of health conditions associated with higher need for SEN provision or disability. We

have not conducted a systematic review. We provide an overview of search terms used for this literature review in the appendix.

- what exactly does the qualitative study look like? (will a questionnaire be developed, how many interviews will be conducted and with whom, how will the interviews be conducted and evaluated, which methodological approach will be used for the analysis, is there ethical approval for this study, especially in view of the fact that (underage) pupils are also to be interviewed? In the abstract you wrote that interviews, focus groups are planned, this information is missing in the methods section. How will you recruit participants?

Thank you, we now provide information about qualitative studies in paragraph 3 of “Methods \ Understanding lived experiences of SEN provision” section (p11) which covers answers to questions above and reads as follows:

“Third, based on results of our national survey of experiences of SEN provision, we will undertake two qualitative studies. The first will use a time-line approach with young people (20-25) and parents (20-25) to gather their experience about the identification, assessment and provision of SEN. Secondly, we will complete nine focus groups with SEN professionals (those working in/closely with education settings, health services, and the local authority) to explore concepts of best practice in SEN and barriers to implementation. There will be three focus groups that will focus on each stage in the process (identification, assessment and provision). We will train and support one parent from our stakeholder group to help facilitate each theme, working with three parents in total. We will recruit participants for both studies from national online survey respondents who have consented to be contacted about further qualitative analyses as part of the survey. Data from each set of stakeholder focus groups and for parents / carers and young people will be analysed separately using a Framework Analysis approach and then compared to look for consensus and differences in views and experiences across informants. We received ethics approval for this study (see Ethics section for details).”

In the Ethics section (p15) we provide additional information about the ethical approvals for the qualitative study: *“Separate ethics approval has been approved for the mixed-methods research (national survey, interviews and focus groups) involving service users (young people and parents) and service providers (PRE:2021.058). Parents consented for their own involvement and also for their child if under the age of 16. Young people aged 16 or over consented and younger children were asked for assent to their participation using a similar process.”*

- How will the online survey be conducted? How will the questionnaire be developed for this purpose, will already established instruments for the survey of health status etc. will be used? How will you recruit participants? You only write about ethics in the abstract.

As the survey has already been completed, we now describe how the survey was designed, disseminated and how data were collected in “Data sources” section (on p7), and we provide some results on numbers of participants. Key findings have not been published yet, but we provide a link to short summaries published on our study website. We also discuss ethics approval in “Ethics” section on p17.

The revised section reads as follows:

“We conducted three online surveys aimed at (1) children and young people, (2) parents/carers and (3) service providers (health, education and local authority professionals) to document variation in local experiences of identification and assessment of need, and provision of SEN interventions. The surveys were developed through a scoping review to identify previous questionnaires and co-designed with stakeholder groups of young people, parents / carers and professionals working in education or health with children who have SEN. Data were collected using REDCap. The surveys were disseminated via social media (Twitter, Instagram, and Facebook), and through professional networks (GOV.UK Notify service, Parent and Carer forums, and stakeholder group contacts). These

networks were used to maximise the recruitment of all three groups from each of the nine regions in England. We received ethics approval for this study (see Ethics section for details).

In total 1,714 participants took part from across England including: 77 young people aged 11-27 years, 772 parents and carers, and 865 service providers (those working in/closely with education settings, the health services, local authorities). Short summaries of the key findings from initial analyses are published on the study website (<https://dev.psychiatry.cam.ac.uk/hope-study-health-outcomes-for-young-people-throughout-education/>) and more detailed papers are in preparation. These data will inform further qualitative components of the study.”

Have the individual studies been pre-registered?

Individual studies have not yet been pre-registered, but we are planning to pre-register study protocols for causal analyses on NIHR Open Research platform (<https://openresearch.nihr.ac.uk/>, see first and last paragraph on p11). We also refer to a pre-print for a protocol for a target trial emulation of SEN provision for a population of children with cleft lip and palate (first paragraph p12). Protocols for other health phenotypes will be pre-registered in due course.

Perhaps the methods, including study design, study population, data extraction/analysis, etc., could be reported for each study rather than for all studies.

Thank you, we have now restructured the protocol. The methods section now covers an overview of different component studies, and study protocols for these will follow.

Reviewer: 2

Dr. Robert French, Cardiff University School of Medicine Comments to the Author:

A very interesting and well-presented protocol, thank you for the opportunity to review this draft.

My primary challenge as a reader was that the broad scope of what is planned within the protocol meant that there was often insufficient information to really grasp the detail of what the researchers were actually going to do for some strands. There was good detail on the context and the data, but it felt less clear on what analysis would take place. In places there was a lot of detail, for example for "exploring variation in recorded SEN", but for the "causal workstream" and "Assessing the Impact of recorded SEN on outcomes" it was less clear. This was partly justified, for example, that the researchers would consult with patients on what were the most relevant outcomes. Also, that reviews of the literature would help inform casual approaches such as potential IVs. However, it still felt a bit unbalanced and more like a grant proposal than a protocol.

Thank you for this comment. We now clarify in the introduction (p7) that this is a protocol for a mixed methods research programme, consisting of multiple component studies and that detailed protocols for each component study will be reported separately.

We have restructured the methods section to provide an overview of shared definitions and planned analyses covered by component studies, including information on:

- The proposed study population – our analyses will focus on children in primary school who were born in England and examine 3 groups of health phenotypes
- Considered study measures – we provide definitions of health and educational outcomes and classification of SEN provision which are shared across studies
- overview of planned analyses for component studies, including descriptive analyses of outcomes across phenotypes, examining variation in SEN provision, mixed-methods work to understand live experiences of SEN provision, and additional detail on how causal analyses will be designed and which methods will be considered (p11-14).

However, there is not enough space to transparently report details of several studies and further study protocols will be written for specific studies. Please see a similar 'umbrella' protocol by Craig et al. <https://bmjopen.bmj.com/content/12/4/e061340>

My suggestion would be to add more detail for wps 3-4 in "study Design". In methods, I would consider trying to be more specific wherever possible, particularly in the first paragraph of

“Assessing the impact of recorded SEN on outcomes”. If more detailed protocols are planned for some of these workstreams it may be worth flagging that here, since that would help justify being less detailed in this protocol?

Thank you for this suggestion. As mentioned above, we now clarify that further specific study protocols will be registered separately. We have restructured all of methods section to provide more detailed information about each component study.

We now added Table 1 with a Roadmap for causal investigations in HOPE, with an exemplar of cleft lip and palate, illustrating the complexities of these analyses.

Abstract: “state-funded hospitals” – is this the right term for the data included in HES?
We now revised this to say “*contacts with National Health Service (NHS) hospitals in England*”.

Intro: “experiencing deprivation” -> “deprivation levels” ?

We have now revised this whole paragraph to read as follows:

“SEN provision across England is widely regarded as inequitable.^{3,5,8,9} The proportion of pupils with SEN support ranged from 7.3% to 17.1% and the proportion with EHCPs from 0.8% to 5.0% across local authorities in 2018/19.⁶ Allocation of SEN provision is associated with a variety of factors. According to a recent report, a key factor determining SEN provision is the school, particularly school’s previous rates of SEN provision, academy status and previous school inspection outcomes.⁵ Other factors include the proportion of academised primary schools and rates of pupils eligible for free school meals at local authority level-and pupil-level factors such as attainment at school entry (age 5), ethnic group, child’s first language and contacts with social care.^{5,8,9} The annual proportion of children with recorded SEN provision has also declined over time, from 20% in 2010 to 14% in 2016. This change seems partly related to the Children and Families Act in 2014 and Special Educational Needs and Disability Code of Practice implemented in 2015, and to reduced funding to local authorities from 2010 (Figure 1).^{3,10}”

Intro: “Social care” data pops up without earlier reference to where it comes from, readers may think this is another dataset

Thank you, we now removed mentions of social care throughout the protocol as this is not covered within the scope of this study (although available in ECHILD database).

Data source: “ECHILD follows” -> “ECHILD includes” ?

We have revised accordingly.

Data source: I may have missed it but it didn’t list who did the linkage

Thank you for spotting this, we now added this information to “methods” section on p6: “Health and education datasets were linked by NHS England using a multi-step deterministic linkage algorithm, described in detail elsewhere.²⁴”

Data source: Q. does “birth admissions” include all births or only those connected with a hospital admission?

We now clarify that birth admissions are hospital admissions resulting in a birth. We therefore include only children born in England in NHS-funded hospitals: “*Nearly all children born in England are born in NHS hospitals (97%) and can be followed from their birth admission through all subsequent hospital contacts.*^{26,27} “

Dissemination: If possible, please set up the GitHub repository (if it not done already) and provide the link to the repository here in the protocol

thank you, we now reference ECHILD GitHub page (<https://github.com/UCL-ECHILD>)

Figure 1: This was helpful for context.

Thank you.

Figure 2 (labelled as Figure 1!): It might just be me, but I did not find this particularly clear or helpful and wonder if the authors could think of ways to make this more easily understandable.

Thank you – we now re-labelled the figure and revised to match planned component studies. However, we can remove this figure if it is not helpful for the protocol.

Figure 3: It may be helpful to add a note to say what is planned (if anything) for 2021/22 data
 ECHILD now covers 2021/22 data and we have updated the figure to include this information.

Great that you have included a glossary.
 Thank you.

Glossary (cf Figure 2): You use "statement" in Figure 2, could that be referenced in your glossary? You may want to add other historical terms like "Action", "Action Plus", to help older readers. Also "Key Stage"

Thank you, we now added these terms to glossary & also include them in the introduction (last paragraph on p4). We also added information about Key Stages in Box 2 and in the glossary.

Glossary (or elsewhere): Make it clear the 'scale' on which SEN is measured eg lowest level is X, highest level is Y. (I think this must have been covered and I've missed it, but noting just in case)

thank you, we have now added a number of study measure definitions for outcomes and SEN provision in Box 2, clarifying we will categorise SEN provision in a descending hierarchy.

VERSION 2 – REVIEW

REVIEWER	Christin , Ellermann Universität Potsdam, Faculty of Health Sciences Brandenburg
REVIEW RETURNED	26-Jul-2023

GENERAL COMMENTS	Thank you for the opportunity to review the updated version of the protocol. In some parts it has become more clearer, but I would still like to suggest more structure. It is still unclear to me how the individual studies/components of the research programme interact with each other. From Figure 2 it is still not clear which research question you are trying to answer with which method and when. There is no need to elaborate on each individual study in detail, as you are going to report each component study elsewhere, but a little more clarification on the interplay/how the individual work packages (WP) come together iteratively would be helpful for the reader, e.g.: Main objective To improve SEN provision and tackle inequitable access to SEN provision a mixed methods study will analyse current situation of SEN provision in England. We will use the following research programme: WP1: Scoping review to identify questionnaires for national online survey and stakeholder meetings with groups of young people, parents / carers and professionals working in education or health with children who have SEN to co-design the surveys (see WP3). WP2: Quantitative analyses of the Education and Child Health Insights from Linked Data (ECHILD) database to describe the current situation of SEN provision in England (e.g., health conditions expected to need SEN provision, prevalence of SEN provision between schools and demographic groups, impact of SEN provision on health and educational outcomes).
--

WP3: National online survey: Online surveys with (1) children and young people, (2) parents/carers and (3) service providers (health, education and local authority professionals) to document variation in local experiences of identification and assessment of need, and provision of SEN interventions.

The three surveys were developed through a scoping review to identify previous questionnaires (see WP1) and co-designed with stakeholder groups of young people, parents / carers and professionals working in education or health with children who have SEN [how was co-design realized?].

WP4: Policy

[will there be a specific policy WP to communicate results of the research programme to policy makers and propose a strategy? If so, how does it look like?]

...

I would suggest to add information on which studies have already been completed in the abstract (state that you will also report some preliminary results in the protocol, if you plan to publish details of the (pre-)studies elsewhere; you could add a section after the methods on preliminary results).

Methods

We will use mixed methods for each component of the study to answer the following research questions:

[What is/are the research question/s/hypotheses to be answered by each WP and how? How does every step inform further steps of the research program /or are they running in parallel (add timeframes for the single components/studies of the research program; which WP are already been completed); add these information in Figure 2]

WP1: Scoping review and stakeholder meetings [description of the research question(s), databases that were searched, search strategy (with reference to the appendix), timeframe, ...]

WP2: Quantitative analyses ...

- [What are the overarching research questions?] (e.g., What kind of "health phenotypes" do exist? What kind of health conditions expected to need SEN provision? What impact has SEN provision on health and educational outcomes? How does SEN provision vary between schools and demographic groups? ...)

WP3: National online survey [What is/are the research question(s)? What is the aim of this WP? (e.g., identification of barriers, involvement of various stakeholders to improve SEN provision, reduce inequity in SEN provision)]

	[WP4: Policy] ... I have the feeling that most of the information in the methods and analysis section belong to the the quantitative analysis of the ECHILD database. Maybe you can make this more clear for the reader. If so, you could report on health phenotypes, how health and education outcomes will be measured, etc. only for this WP. Please find below some comments and suggestions: Page 7, line 42: Study population: health phenotypes I would suggest to add this into a section on preliminary results, where you report on the studies that have already been done (e.g., WP1+2?).
--	---

REVIEWER	French, Robert Cardiff University School of Medicine, Medicine
REVIEW RETURNED	30-Jul-2023

GENERAL COMMENTS	Thanks to the authors for addressing our comments and for providing the file summarising responses to each of the issues. There was a lot of new text, which possibly went beyond the requested edits, although this is understandable given how the work has progressed, this made it challenging to do the second review, so apologies if I have commented on aspects that have been addressed etc. I have covered my previous comments first, then new comments about the whole paper in order. My bottom line is that the authors need to clearly spell out to the reader what specific projects are under the protocol umbrella, and then throughout to focus on the things that are common, direct the reader to the new protocols for the study specific information, and provide study specific details only where it is helpful to understand the whole. I realise drafting an umbrella protocol is a very challenging task, particularly as things are developing all the time, but this will be a very useful and important document for people to understand how this incredible dataset can be used, so it is worth the effort to get it right. Most of these changes are fairly superficial and so should be quick to do on a single read through, with my major comment probably more easily tackled with judicious cutting/editing/relegating to an appendix, rather than lots of new text. Where comments are listed as suggestions, I would still expect the authors to acknowledge the comment even if they choose not to act upon it. Thanks again for the opportunity to review a very exciting paper. I look forward to seeing the results from the studies. Best wishes, . Previous comments 1: My key concern [comment 1], which was shared by the other reviewer, about helping the reader navigate the breadth of the study, has not been fully addressed. Though there is a great deal of new text, I still feel the paper falls short in terms of the balance between the component studies and how this impacts on the overall readability of the paper. Comment 2 The fact that this is an 'umbrella' protocol needs to be clearly signposted throughout the document, but particularly
---

	towards the top. I am not going to give all of specific changes, I have included a couple of examples below, but would like the authors to check that this is made clear throughout the document.  • In the abstract ‘In this mixed methods study, we” suggests a single study, this would be the first place to emphasise the umbrella protocol’. • At some point I would like to see the authors really spell out what is under the umbrella, e.g. “the research programme includes three studies, study 1 will do X, study 2 will do y, study x will do Z,”, then offer more detail for each, make it absolutely clear for the reader where something is common and where it is study specific. • I thought the new para at the end of the intro is really helpful to understand the scope, but this part “The programme integrates quantitative analyses of the Education and Child Health Insights from Linked Data (ECHILD) database (see data resources) with mixed quantitative and qualitative methods to understand variation in identification, assessment and provision for SEN, and how these processes are experienced by families. We will explore which causal inference methods can be used to provide valid evidence of the impact of SEN provision on health and education outcomes, and in which contexts.” Could be expanded here to cover the comment above. • (Also for some readers this new paragraph on scope might come too late to make sense of the literature which is a bit disjointed given the scope of the research studies. One option the authors might consider is to have the Introduction section (i) defining SEN first, then (ii) the new para at the end of the intro AND shift all the literature to a new ‘Background’ section for which the reader will then have a sense of what the different research elements are which underpin the literature. This is just a suggestion, rather than a direction. I realise this would take some work, and the authors may argue that the literature ‘sets up’ the research and hence justifies putting that para at the end of the intro after the literature, so just including it for the authors as a suggestion.) • The analysis plan still feels very unbalanced. Given the umbrella nature of the protocol, it has to focus on those aspects which are common to the studies, and not go off on too many tangents specific to one study. I would suggest the text on the individual studies in the analysis plan are significantly reduced and the text put aside for use in the study specific protocols. I would perhaps suggest one long paragraph for each study in the analysis plan, and consistent information provided for each study (including where the reader can expect to find more detail in due course), currently you are specifying different types and levels of information for each study which makes it hard to read. // I realise this slightly contradicts the other reviewers comments in terms of the detail required, the key thing for me is achieving balance between the different studies. I am satisfied with responses to [comments 3, 4, 5] [Comments 6] birth admissions – A very minor point, but I still think this is not explaining this important point clearly enough, perhaps, go on to say that those without hospital admissions for birth would include those born overseas, born at home, born in other settings (prisons?) etc. (if that reflects the facts). Though happy for the authors to address this in another way, I just want the reader to be clear on who has mother-child link. I am satisfied with responses to [comments 7, 8]
--	---

	Comment 9 – Figure 2. I still feel this important figure is falling short. I would not insist on any specific changes, but emphasise that this is a real opportunity to present the whole piece in a clear way, to put a really memorable image in the mind of the reader of the different parts of the study and the connections between them. This also comes back to comment 1 about how you communicate to the reader that is an umbrella protocol. I'd encourage the authors to think again about how they could improve it, perhaps replacing some of the bullet text with images, perhaps putting more of the text from the figure into the text description in the manuscript that refers to this figure, .maybe splitting out some of the boxes? I am satisfied with responses to [comments 10, 11, 12, 13] New comments: Reading afresh I would make a few suggestions to the authors, I did not want to be proscriptive, and none of these are dealbreakers. Abstract: I wonder if this “We will triangulate results to generate evidence to inform policy.” really reflects the text in the main document, maybe a better form of words, emphasising how the different types of evidence might provide more robust support for policy change // or whether policy comments are even required in the abstract of this paper which does not directly focus on policy? Abstract: Similarly, while “We are working with these stakeholders to help us interpret, frame and communicate our findings to policy makers, health and education services and families in order to promote translation of our findings into practice.” While this is probably true and a good thing, does it really reflect the contents of the manuscript, and so might be more focused on how PPIE supported the development of the research questions , define variables etc. IE the parts that form the bulk of the paper? Strengths and limitations: These all focus on the dataset, which is not really the focus of the manuscript (i.e. the different research studies) For Box 1 – have these definitions been the case for all the data they are using (i.e. back to 1997) or just the most recent years? As a minimum the authors might want to indicate the time period for which Box 1 is correct in the box title, or something like ‘at the time of publication’ if it hard to ascertain the exact dates. // A related thought is whether there is a danger that the older SEN data is misrepresented in quantitative studies that treat the categories as they are interpreted now? This comment about representing the history of all variables (not just SEN) is going to be a common challenge for the studies which use the data This text “As these are deidentified data, no consent is required for use” is true and although they do state at the start of the para that this is for the processed echild data, it does not sit well with me, given the processing of CPI earlier in the linkage process. In my opinion consent is irrelevant to this part and I would cut this sentence and only include discussion of consent where I was talking about CLDoC and/or consent for parts of the qual study. Fig 3 goes back to 2001 for education data, Fig 1 goes from 2003, is this worth a footnote comment in figure 1? In the table of key stages, is there a footnote explaining ‘reception’, foundation stage etc.? I would flag to the editors that Figure 1 may need some work to make small text and images readable in the published version.
--	--

VERSION 2 – AUTHOR RESPONSE

Reviewer 1:

Thank you for the opportunity to review the updated version of the protocol. In some parts it has become more clearer, but I would still like to suggest more structure. [Comment 1:] It is still unclear to me how the individual studies/components of the research programme interact with each other.

Response 1: Thank you, we made further revisions to the manuscript to clarify that this is an umbrella protocol describing four parallel work packages (and we briefly describe component studies contributing to each WP), and how these WPs are linked. This is now also illustrated in Figure 2. In short:

- In WP1 we defined a range of “health phenotypes”, that is health conditions captured in hospital records that are expected to need SEN provision in primary school. As next step we explore how health and education outcomes vary for children with different health phenotypes compared to unaffected peers.
- In WP2, we describe how child, social and area-level factors affect variation in SEN provision within phenotypes.
- In WP3, we apply a range of causal inference methods to address confounding factors (informed by WP2) and possible selection bias to assess the impact of SEN provision on outcomes for children with selected health phenotypes (defined in WP1), also considering timing, duration and level of provision.
- In WP4 we review local policies to describe barriers to and good practice for SEN provision, and synthesise findings from surveys, interviews and focus groups of service users and providers to understand who is identified, assessed and provided with SEN intervention and what factors influence their experience.

Changes made throughout the manuscript include:

- We briefly describe each WP and how they are linked in methods section of the abstract & introduction (paragraphs 2-3 on page 5);
- We describe data sources and study measures shared between WP1-3 in methods separately to data sources used for mixed-method studies in WP4;
- We clarified which analyses are part of which WP using appropriate headings in “analysis plan” section
- We mapped WPs to research questions in Figure 2, and we flag which WPs use ECHILD

[Comment 2:] From Figure 2 it is still not clear which research question you are trying to answer with which method and when. There is no need to elaborate on each individual study in detail, as you are going to report each component study elsewhere, but a little more clarification on the interplay/how the individual work packages (WP) come together iteratively would be helpful for the reader, e.g.:

Main objective

To improve SEN provision and tackle inequitable access to SEN provision a mixed methods study will analyse current situation of SEN provision in England. We will use the following research programme:

WP1: Scoping review to identify questionnaires for national online survey and **stakeholder meetings** with groups of young people, parents / carers and professionals working in education or health with children who have SEN to co-design the surveys (**see WP3**).

WP2: Quantitative analyses of the Education and Child Health Insights from Linked Data (ECHILD) database to describe the current situation of SEN provision in England (e.g., health conditions expected to need SEN provision, prevalence of SEN provision between schools and demographic groups, impact of SEN provision on health and educational outcomes).

WP3: National online survey: Online surveys with (1) children and young people, (2) parents/carers and (3) service providers (health, education and local authority professionals) to document variation in local experiences of identification and assessment of need, and provision of SEN interventions.

R2: We simplified Figure 2 in response to reviewer 2, and we mapped each work package & linked research question. We also clarify which WPs are using ECHILD.

The three surveys were developed through a scoping review to identify previous questionnaires (see WP1) and co-designed with stakeholder groups of young people, parents / carers and professionals working in education or health with children who have SEN [Comment 3: how was co-design realized?].

R3: Thank you for this question. Detailed information on survey design and findings will be published separately, although we now provide additional brief overview on page 9. Revised text now reads: *“Detailed information on survey design and findings will be published separately. In brief, the surveys were developed through a scoping review to identify previous questionnaires and co-designed with stakeholder groups of young people, parents/carers and professionals working in education or health with children who have SEN. Each survey underwent three rounds of extensive piloting with the respective advisory groups from the HOPE study. This helped to: i) ensure that the questions and response options matched the lived experience of participants, ii) test accessibility and usability for respective participants, iii) ensure no technical difficulties existed when completing the survey across multiple device types (e.g smartphone, laptop).”*

WP4: Policy

[Comment 4: will there be a specific policy WP to communicate results of the research programme to policy makers and propose a strategy? If so, how does it look like?]

R4: There will not be a specific work package related to policy. Instead, within each work package we will disseminate our findings to wide range of stakeholders (academics, policy analysts in government department, and service users and providers). Please see dissemination section (p14) for details

Comment 5: I would suggest to add information on which studies have already been completed in the abstract (state that you will also report some preliminary results in the protocol, if you plan to publish details of the (pre-)studies elsewhere; you could add a section after the methods on preliminary results).

R5: Due to word limit of the abstract (300 words), there is no scope to add more detailed information on completed / ongoing work in the abstract, but we now explain that we already defined health phenotypes (other studies are still ongoing).

We do not intend to include preliminary findings in this protocol as these will be published elsewhere. We only include overview of numbers in the study cohort used across WP1-3 and number of people who completed the survey to provide an overview of data sources.

Methods

We will use mixed methods for each component of the study to answer the following research questions:

[Comment 6: What is/are the research question/s/hypotheses to be answered by each WP and how? How does every step inform further steps of the research program /or are they running in parallel (add timeframes for the single components/studies of the research program; which WP are already been completed); add these information in Figure 2]

Response 6: Please see responses 1-2 detailing research questions covered by each work package. All work packages run in parallel, we have frequent project meetings to update the team on findings up to date and inform parallel work. We clarify that this work runs in parallel in paragraph 2 on p5.

WP1: Scoping review and stakeholder meetings [Comment 7: description of the research question(s), databases that were searched, search strategy (with reference to the appendix), timeframe, ...]

Response 7: We feel that this is too detailed for this umbrella protocol describing an overview of work carried out across multiple packages, especially as this is only one element of the HOPE research programme. We provided additional detail on survey design and we flag that detailed description of

how the survey was designed will be described in paper with results. Similarly, we flag that detailed phenotyping paper is in preparation in first paragraph of “Neurodisability and other high-risk conditions section” (last sentence on p7).

WP2: Quantitative analyses ...

- [Comment 8: What are the overarching research questions?] (e.g., What kind of “health phenotypes” do exist? What kind of health conditions expected to need SEN provision? What impact has SEN provision on health and educational outcomes? How does SEN provision vary between schools and demographic groups? ...)

Response 8: We propose proof of concept analyses of ECHILD to address two core research questions:

- i) which factors contribute to variation in SEN provision in England?
- ii) what is the impact of SEN provision on health and education outcomes?

We state these research questions in the introduction (paragraph 1, page 5). All work packages will support answering these research questions (see response to comments 1 & 2). For example, WP1 will examine whether there are differences in health and education outcomes for children with different health phenotypes, WP2 will examine factors associated with variation, both will inform design of causal studies looking at impact of SEN provision (WP3).

WP3: National online survey [What is/are the research question(s)? What is the aim of this WP? (e.g., identification of barriers, involvement of various stakeholders to improve SEN provision, reduce inequity in SEN provision)]

Response 9: thank you, we now explain that national online survey results were designed to document variation in local experiences of identification and assessment of need, and provision of SEN intervention (paragraph 1 on p9).

[WP4: Policy] ...

I have the feeling that most of the information in the methods and analysis section belong to the quantitative analysis of the ECHILD database. Maybe you can make this more clear for the reader. If so, you could report on health phenotypes, how health and education outcomes will be measured, etc. only for this WP.

Response 10: this is correct, as the HOPE research programme proposes novel proof-of-concept analyses of the ECHILD database in WP1-3, contextualised with findings from mixed methods studies in WP4 (surveys, interviews and focus groups with service users and providers, and document analyses).

We now clarify this in the manuscript:

- we have changed the title to emphasise that analysis of administrative data is at the core of the HOPE study
- we describe shared data sources and study measures for WP1-3 in one section (paged 5-8)
- We visualise the four WPs in Figure 2, separating WPs based on ECHILD
- We made additional changes to the abstract (see response 1)

Please find below some comments and suggestions:

Page 7, line 42: Study population: health phenotypes

I would suggest to add this into a section on preliminary results, where you report on the studies that have already been done (e.g., WP1+2?).

Response 11: All of the studies are ongoing, and these are just preliminary results to describe the study populations used in WP1-3. Developing study cohorts and phenotypes serves as groundwork for all further work and we felt that it is useful for reader to get a sense of population sizes that we are likely to work with.

Page 6, Line 43-44: Compared to their peers, children with SEN provision have higher rates of chronic physical and mental health conditions and hospitalisations, and have lower self-reported wellbeing.”

Do you mean “children with SEN” or “children needing SEN provision”?

R12: As the cited studies used school recorded SEN provision we feel that “children with SEN provision” is more accurate than “children needing SEN provision”.

Page 7, Line 7-8: “...evidence is needed to guide effective intervention to groups of children who are most likely to benefit.”

I would suggest to write “...evidence is needed to provide effective interventions for groups of children who have been less likely to benefit. This will help to reduce inequalities in the provision of SEN services.”

R13: We do not agree with this suggestion as it changes the meaning of the sentence.

Page 13, Table 1: you wrote on page 7 that table 1 will be in the appendix. I believe that this is not the correct reference (are you referring to Figure 1 on page 7?). Please check again (the right reference for Table 1 is on page 13, line 12). I would suggest including Table 1 in the appendix.

R14: Thank you, we were referring there to *Appendix Table 1* (see page 4 of appendix). We now also moved table 1 to the appendix (now listed as *Appendix Table 2*).

Reviewer 2:

Dear Editors & Authors,

Thanks to the authors for addressing our comments and for providing the file summarising responses to each of the issues. There was a lot of new text, which possibly went beyond the requested edits, although this is understandable given how the work has progressed, this made it challenging to do the second review, so apologies if I have commented on aspects that have been addressed etc. I have covered my previous comments first, then new comments about the whole paper in order.

My bottom line is that the authors need to clearly spell out to the reader what specific projects are under the protocol umbrella, and then throughout to focus on the things that are common, direct the reader to the new protocols for the study specific information, and provide study specific details only where it is helpful to understand the whole. I realise drafting an umbrella protocol is a very challenging task, particularly as things are developing all the time, but this will be a very useful and important document for people to understand how this incredible dataset can be used, so it is worth the effort to get it right.

Most of these changes are fairly superficial and so should be quick to do on a single read through, with my major comment probably more easily tackled with judicious cutting/editing/relegating to an appendix, rather than lots of new text. Where comments are listed as suggestions, I would still expect the authors to acknowledge the comment even if they choose not to act upon it.

Thanks again for the opportunity to review a very exciting paper. I look forward to seeing the results from the studies.

Best wishes,

Previous comments 1:

My key concern, which was shared by the other reviewer, about helping the reader navigate the breadth of the study, has not been fully addressed. Though there is a great deal of new text, I still feel the paper falls short in terms of the balance between the component studies and how this impacts on the overall readability of the paper.

Response 15: Thank you, we made further revisions to help navigate work ongoing and planned in each work package, and how they come together. We now clarify that, in WP1 we defined a range of “health phenotypes”, that is health conditions captured in hospital records that are expected to need SEN provision in primary school. Next, we will explore how health and education outcomes vary for children with different health phenotypes and compared to unaffected peers. In WP2, we describe how child, social and area-level factors affect variation in SEN provision within phenotypes. In WP3, we apply a range of causal inference methods to address confounding factors (informed by WP2) and possible selection bias to assess the impact of SEN provision on outcomes for children with selected health phenotypes (defined in WP1), also considering timing, duration and level of provision. In WP4 we review local policies to describe barriers to and good practice for SEN provision, and synthesise findings from surveys, interviews and focus groups of service users and providers to understand who is identified, assessed and provided with SEN intervention and what factors influence their experience.

Changes made throughout the manuscript include:

- We briefly describe each WP and how they are linked in methods section of the abstract & introduction (paragraphs 2-3 on page 5);
- We describe data sources and study measures shared between WP1-3 in methods separately to data sources used for mixed-method studies in WP4;
- We clarified which analyses are part of which WP using appropriate headings in “analysis plan” section
- We mapped WPs to research questions in Figure 2, and we flag which WPs use ECHILD
- We also shortened description of analyses planned for WP3 (causal inference) and moved Table 1 to the appendix to provide more balance between component studies.

The fact that this is an ‘umbrella’ protocol needs to be clearly signposted throughout the document, but particularly towards the top. I am not going to give all of specific changes, I have included a couple of examples below, but would like the authors to check that this is made clear throughout the document.

Response 16: thank you, we now clarify that this is an “umbrella” protocol throughout the protocol: we added it to the title, abstract, and paragraph 2 of p7 of the introduction.

In the abstract ‘In this mixed methods study, we” suggests a single study, this would be the first place to emphasise the umbrella protocol’.

Response 17: Thank you for this suggestion. We revised abstract (in part to meet word count requirements) and the first sentence now reads *“The HOPE research programme uses administrative data from the Education and Health Insights from Linked Data – ECHILD – which contains data from all state schools, and contacts with NHS hospitals in England, to explore variation in SEN provision and its impact on health and education outcomes. This umbrella protocol sets out analyses across four work packages (WP).”*

At some point I would like to see the authors really spell out what is under the umbrella, e.g. “the research programme includes three studies, study 1 will do X, study 2 will do y, study x will do Z,”, then offer more detail for each, make it absolutely clear for the reader where something is common and where it is study specific.

Response 18: Thank you for this suggestion. We now clarify that this umbrella protocol sets out planned and ongoing work across four parallel work packages and summarise objectives of each WP in the abstract, introduction and methods. See response 15 for details.

I thought the new para at the end of the intro is really helpful to understand the scope, but this part *“The programme integrates quantitative analyses of the Education and Child Health Insights from Linked Data (ECHILD) database (see data resources) with mixed quantitative and qualitative methods to understand variation in identification, assessment and provision for SEN, and how these processes are experienced by families. We will explore which causal inference methods can be used to provide valid evidence of the impact of SEN provision on health and education outcomes, and in which contexts.”* Could be expanded here to cover the comment above.

R19: Thank you for this suggestion. We have revised the end of the introduction (paragraphs 2-3 on page 5) to clarify work carried out in each work package.

(Also for some readers this new paragraph on scope might come too late to make sense of the literature which is a bit disjointed given the scope of the research studies. One option the authors might consider is to have the Introduction section (i) defining SEN first, then (ii) the new para at the end of the intro AND shift all the literature to a new 'Background' section for which the reader will then have a sense of what the different research elements are which underpin the literature. This is just a suggestion, rather than a direction. I realise this would take some work, and the authors may argue that the literature 'sets up' the research and hence justifies putting that para at the end of the intro after the literature, so just including it for the authors as a suggestion.)

R20: Thank you for this suggestion. We have substantially cut down the introduction and moved more detailed information on SEN provision in England to Appendix 1.

The analysis plan still feels very unbalanced. Given the umbrella nature of the protocol, it has to focus on those aspects which are common to the studies, and not go off on too many tangents specific to one study. I would suggest the text on the individual studies in the analysis plan are significantly reduced and the text put aside for use in the study specific protocols. I would perhaps suggest one long paragraph for each study in the analysis plan, and consistent information provided for each study (including where the reader can expect to find more detail in due course), currently you are specifying different types and levels of information for each study which makes it hard to read. // I realise this slightly contradicts the other reviewers comments in terms of the detail required, the key thing for me is achieving balance between the different studies.

R21: Thank you, we made further revisions to clarify what analyses are planned for each work package. We clarify that we describe shared definitions for analyses of ECHILD used across WP1-3. We also shortened description of analyses planned for WP3 (causal inference, analyses described on pages 10-11) and moved Table 1 to the appendix to provide more balance between component studies. We group all mixed-methods analyses in WP4.

I am satisfied with responses to [comments 3, 4, 5]

[Comments 6] birth admissions – A very minor point, but I still think this is not explaining this important point clearly enough, perhaps, go on to say that those without hospital admissions for birth would include those born overseas, born at home, born in other settings (prisons?) etc. (if that reflects the facts). Though happy for the authors to address this in another way, I just want the reader to be clear on who has mother-child link.

R22: We added clarification to ECHILD Database section to say "Nearly all children born in England are born in NHS hospitals (97%) but HES excludes births in private hospitals or at home" (last paragraph, p5). We do not mention mother-child link in the protocol, as we are not relying on mother-baby linked data in the HOPE study.

I am satisfied with responses to [comments 7, 8]

Comment 9 – Figure 2. I still feel this important figure is falling short. I would not insist on any specific changes, but emphasise that this is a real opportunity to present the whole piece in a clear way, to put a really memorable image in the mind of the reader of the different parts of the study and the connections between them. This also comes back to comment 1 about how you communicate to the reader that is an umbrella protocol. I'd encourage the authors to think again about how they could improve it, perhaps replacing some of the bullet text with images, perhaps putting more of the text from the figure into the text description in the manuscript that refers to this figure, .maybe splitting out some of the boxes?

R23: thank you, we now simplified Figure 2 (taking some of the text away) and we added information on where each work package fits in. We also added two different coloured backgrounds to separate WPs using ECHILD and WP4 which uses mixed-methods to contextualise the findings from ECHILD.

I am satisfied with responses to [comments 10, 11, 12, 13]

New comments:

Reading afresh I would make a few suggestions to the authors, I did not want to be proscriptive, and none of these are dealbreakers.

- Abstract: I wonder if this “We will triangulate results to generate evidence to inform policy.” really reflects the text in the main document, maybe a better form of words, emphasising how the different types of evidence might provide more robust support for policy change // or whether policy comments are even required in the abstract of this paper which does not directly focus on policy?
R24: Thanks for making this point. We have changed the last sentence of the abstract to clarify that triangulation of findings will inform interpretation of findings for policy, practice and families and methods for future evaluation.
- Abstract: Similarly, while “We are working with these stakeholders to help us interpret, frame and communicate our findings to policy makers, health and education services and families in order to promote translation of our findings into practice.” While this is probably true and a good thing, does it really reflect the contents of the manuscript, and so might be more focused on how PPIE supported the development of the research questions, define variables etc. IE the parts that form the bulk of the paper?
R25: We agree and have revised this point as “*These stakeholders will contribute to the design, interpretation and communication of findings*”
- Strengths and limitations: These all focus on the dataset, which is not really the focus of the manuscript (i.e. the different research studies)
R26: thank you, we have revised the S&L to include information about causal methods and mixed methods used in the HOPE programme, as well as ECHILD.
- For Box 1 – have these definitions been the case for all the data they are using (i.e. back to 1997) or just the most recent years? As a minimum the authors might want to indicate the time period for which Box 1 is correct in the box title, or something like ‘at the time of publication’ if it hard to ascertain the exact dates. // A related thought is whether there is a danger that the older SEN data is misrepresented in quantitative studies that treat the categories as they are interpreted now? This comment about representing the history of all variables (not just SEN) is going to be a common challenge for the studies which use the data
R27: thank you, we now added a footnote to Box 1 (not that it was moved to the appendix) explaining that the sub-categories changed in 2014/15 following reforms to SEN system: “Social, emotional and mental health difficulties” were introduced in 2014/15, while “Behaviour, Emotional & Social Difficulties” were removed.

We agree about challenges in using the data where SEN provision changed over time. We will consider impact of these changes when designing studies (e.g. when selecting time frame for follow-up) and report changes in SEN provision over time. We have added a paragraph about these changes on page 10 (last paragraph of “Recorded SEN provision” section).
- This text “As these are deidentified data, no consent is required for use” is true and although they do state at the start of the para that this is for the processed echild data, it does not sit well with me, given the processing of CPI earlier in the linkage process. In my opinion consent is irrelevant to this part and I would cut this sentence and only include discussion of consent where I was talking about CLDoC and/or consent for parts of the qual study.
R28: Thank you for this comment, we now deleted that sentence.
- Fig 3 goes back to 2001 for education data, Fig 1 goes from 2003, is this worth a footnote comment in figure 1?
R29: Figure 1 is based on publicly available data from DfE website & SEN disability review, rather than from NPD data that we used. Figure 3 shows NPD data available in ECHILD. We

were not able to find published data going back to 2001, but we now added a data source to Figure 1 and a clarification that it's based on published figures from DfE.

- In the table of key stages, is there a footnote explaining 'reception', foundation stage etc.?
R30: we now added Early Years Foundation Stage to the description of stages of national curriculum in the appendix and we refer to this appendix in Box 1 now
- I would flag to the editors that Figure 1 may need some work to make small text and images readable in the published version.